# Rumen Microbiota Predicts Feed Efficiency of Primiparous Nordic Red Dairy Cows

**DOI:** 10.3390/microorganisms11051116

**Published:** 2023-04-25

**Authors:** Miika Tapio, Daniel Fischer, Päivi Mäntysaari, Ilma Tapio

**Affiliations:** 1Genomics and Breeding, Production Systems, Natural Resources Institute Finland (Luke), 31600 Jokioinen, Finland; miika.tapio@luke.fi; 2Applied Statistical Methods, Natural Resources, Natural Resources Institute Finland (Luke), 31600 Jokioinen, Finland; daniel.fischer@luke.fi; 3Animal Nutrition, Production Systems, Natural Resources Institute Finland (Luke), 31600 Jokioinen, Finland; paivi.mantysaari@luke.fi

**Keywords:** ruminants, feed efficiency, rumen bacteria, metagenomics, prediction

## Abstract

Efficient feed utilization in dairy cows is crucial for economic and environmental reasons. The rumen microbiota plays a significant role in feed efficiency, but studies utilizing microbial data to predict host phenotype are limited. In this study, 87 primiparous Nordic Red dairy cows were ranked for feed efficiency during their early lactation based on residual energy intake, and the rumen liquid microbial ecosystem was subsequently evaluated using 16S rRNA amplicon and metagenome sequencing. The study used amplicon data to build an extreme gradient boosting model, demonstrating that taxonomic microbial variation can predict efficiency (r_test_ = 0.55). Prediction interpreters and microbial network revealed that predictions were based on microbial consortia and the efficient animals had more of the highly interacting microbes and consortia. Rumen metagenome data was used to evaluate carbohydrate-active enzymes and metabolic pathway differences between efficiency phenotypes. The study showed that an efficient rumen had a higher abundance of glycoside hydrolases, while an inefficient rumen had more glycosyl transferases. Enrichment of metabolic pathways was observed in the inefficient group, while efficient animals emphasized bacterial environmental sensing and motility over microbial growth. The results suggest that inter-kingdom interactions should be further analyzed to understand their association with the feed efficiency of animals.

## 1. Introduction

Improving feed efficiency of dairy cows has become increasingly important for both economic and environmental reasons. Feed accounts for a significant portion of total milk production costs; therefore, improving feed utilization could enhance the profitability of milk production. Furthermore, improving feed efficiency in dairy production reduces nutrient and greenhouse gas emissions and decreases the need for land and resources to produce feed [1,2,3]. Several studies have demonstrated genetic variation among cows in their feed efficiency [4,5,6]. This finding suggests that feed efficiency should be included in the breeding goals of dairy cattle. However, choosing suitable feed efficiency traits can be challenging [7,8]. Efficiency can be defined as ratio of (energy corrected) milk yield to feed (or energy) intake [3]. These measures can be problematic because they favor dairy cows with long and deep energy deficits in early lactation [3,9]. Alternative measures estimate net feed efficiency or metabolic efficiency of the cow. It is defined as the difference between a cow’s actual feed (or energy) intake and its expected feed (or energy) requirements [3]. Residual energy intake (REI) is calculated as the residual from a regression model of energy intake on energy sinks for milk production, maintenance, and body weight change [9], where cows with low REI values indicate more efficient energy utilization.

There is growing interest in understanding the associations between animal performance efficiency and the rumen microbiota. This interest is not surprising, given that symbiotic microorganisms in the rumen are responsible for feed digestion and production of volatile fatty acids (VFA). The host absorbs and utilizes these VFA as an energy source, while the microbial cells, after being washed out of the rumen, become a source of protein for the animal. With the development of high throughput sequencing techniques that allow screening of environmental samples, multiple studies have demonstrated that rumen microbiota is affected by diet [10,11], or breed [12,13,14], and can be associated with methane emissions [15,16,17] or feed efficiency [18,19]. For instance, Jewell et al. [18] studied Holstein cows over a 2-lactation period and observed that inefficient cows had a higher abundance of *Butyrivibrio,* while efficient animals had more *Coprococcus*. Similarly, higher abundance of *Megasphaera elsdenii* and *Coprococcus* were linked with animal efficiency in [19]. Differential abundance of *Methanobrevibacter* species in efficient and inefficient Texel cross Scottish Blackface sheep [20], and Holstein cows [19] have been reported as well. Paz et al. [21] suggested that approximately 20% of the variation in feed efficiency traits could be explained by the rumen microbiome in beef cattle. However, it is difficult to draw more general conclusions on the associations between individual microbial taxa and animal efficiency phenotype from individual studies due to diet having a significant effect on the composition of the rumen microbiome. Recent larger cohort studies suggest that, rather than abundances of individual taxa, the lower taxonomic diversity of rumen bacterial ecosystem, as well as less diverse functional potential of the rumen microbiome [19,22], or differences in microbial functional capacities [13,23] could explain differences between efficient and inefficient animals. All these observations suggest that rumen microbial function is one of the factors that influences animal feed efficiency.

Research exploring the relationship between microbiota and feed efficiency is often constrained by small sample size. These limitations not only reduce statistical power, but also impede the utilization of more advanced machine learning methods that could provide predictions and more comprehensive insights. Several studies have therefore attempted to identify microbial biomarkers that could be used to identify more efficient animals without testing predictive power [20,24,25,26]. However, Clemmons et al. [27] identified biomarkers for high and low feed efficient steers and reached acceptable predictive classification accuracy based on biochemical and taxonomical markers using the random forest algorithm. Furthermore, Delgado et al. [28] used whole metagenome sequencing with enrichment analysis to identify associated microbial contigs and identified strong correlations within the study population and used LASSO regression to predict feed efficiency in an unrelated sample. While regression, linear discriminant analysis, and random forests are considered the most robust machine learning methods, gradient boosting and neural networks are recognized for their greater precision when tuned carefully [29,30]. Although these methods have been successfully applied in transcriptome analysis [31,32], they are underutilized in rumen microbiome research.

The aims of this study were to explore the association between the rumen microbiome and feed efficiency in a cohort of 87 Nordic Red dairy cows and to elucidate on the usefulness of rumen microbiota in predicting feed efficiency phenotype. We were able to predict feed efficiency trait based on rumen microbial composition and were able to link these to variation in microbial processes in dairy cow rumen.

## 2. Materials and Methods

### 2.1. Animals, Experimental Design, and Diet

All experimental procedures were approved by the Project Authorization Board (Regional Administrative Agency for Southern Finland, Hämeenlinna, Finland; ESAVI/9532/2018 and ESAVI/11796/2014) in accordance with the guidelines established by the European Community Council Directive 86/609/EEC. Eighty-seven primiparous Nordic Red dairy cows from Luke research barn in Finland were studied for their energy use efficiency. Animal data and sample collection were conducted during the 2018–2020 period, excluding summertime when cows were on pasture. Twenty-one cows were enrolled in the study in 2018, twenty-eight in 2019, and thirty-eight animals in 2020, respectively.

Cows were housed in a tie-stall barn and were fed grass silage and concentrate mix, provided through the feeding stations, as described by [33]. The mean composition of grass silage and concentrate mix during data collection period are presented in Appendix A. The proportion of concentrate in the diet was adjusted based on the stage of lactation and the digestibility of the grass silage. The mean feed and nutrient intake of the cows from calving to rumen sampling during the different years are given in Appendix A.

### 2.2. Sampling and Data Recording

The body weight (kg) of the cows was recorded using a walk-through static scale (Pellon Group Oy, Ylihärmä, Finland) on their return from milking. Daily milk yield (kg/day), milk composition (%), and dry matter intake (kg/day) were monitored as described by [33]. All feeds were sampled and analyzed to obtain metabolizable energy and nutritional values for the feeds. Rumen liquid samples from each cow were collected when animals were on average 148 (SD 18) days in milk (DIM). Rumen liquid (ca. 500 mL) was collected ca. 3 h after morning feeding via esophagus using Ruminator device (Profs Products, Wittybreut, Germany). Immediately after collection, rumen liquid was strained through two layers of cheesecloth, a subsample of 5 mL was mixed with 0.5 mL of saturated HgCl_2_ and 2.0 mL of 1 M NaOH solutions and stored at −20 °C for subsequent volatile fatty acid (VFA) analysis by gas chromatography as described by [34]. For rumen microbial community determination, rumen liquid was aliquoted into 2 mL tubes, snap frozen in dry ice, and stored at −80 °C until DNA extraction.

### 2.3. DNA Extraction, Sequencing

Total DNA was extracted from 0.5 mL of rumen liquid using DNeasy Power Soil Pro kit (Qiagen, Hilden, Germany) and following manufacturer’s protocol with initial homogenization and mechanical cell disruption performed by bead beating at 6 m/s × 1 min × 3 times in FastPrep (MP Biomedicals, Solon, OH, USA). Rumen bacterial community composition of all 87 cows was determined using universal primers 515F and 806R [35] for 16S rRNA gene V4 region amplicon sequencing. Sequencing library was prepared as described by [36] and sequenced in Finnish Functional Genomics Centre (Turku, Finland) on Illumina MiSeq platform by using 2 × 300 bp chemistry. For shotgun-metagenome sequencing 32 samples were shortlisted based on REI values to represent 11 lowest (L), 10 middle (M), and 11 highest (H) samples maximizing among group differences. The M group had the five highest of the lower median half and the five lowest of the high median half, respectively. For all samples 2 × 150 bp libraries were prepared using the NexteraXT DNA Library Preparation Kit (Illumina, San Diego, CA, USA), following the manufacturer’s guidelines. Sequencing was performed at the LaBSSAH-CIBIO Next Generation Sequencing Facility of the University of Trento on the Illumina NovaSeq 6000 platform (Illumina Inc., San Diego, CA, USA) following manufacturer’s protocols.

### 2.4. Sequencing Data Processing

16S rRNA gene amplicons. Demultiplexing of sequences, adapter removal, and sorting sequences by barcode were performed by the sequencing center. Sequencing data were processed using Qiime 2 [37]. Briefly, quality control, filtering of chimeric reads, and clustering of bacterial sequences into amplicon sequence variants (ASV) were performed using DADA2 [38]. ASVs with less than 10 reads in total were removed. Bacterial ASV taxonomy was assigned using the Silva 138 database [39]. After quality filtering, 2,635,661 reads (mean 30,294/animal) remained for subsequent analyses.

Shotgun-metagenomes. Sequencing resulted in 10.5–140.7 GB per sample. Raw lane-wise sequencing reads were concatenated per sample and quality trimmed with fastp [40], using a hard trimming of 12bp at the 5′-end, automatic adapter removal and a quality filtering of ≥Q15. Quality trimmed fastq-files were then concatenated into a single file-pair and MEGAHIT [41] was used to construct the metagenome assembly, setting the minimum k-mer size to 31. The created metagenome assembly contained approximately 28.3 MM contigs, with an average length of 629bp and a N50 length of 742bp. In total, 37.1 MM genes on the assembled contigs were predicted with Prodigal [42] and, from them, 24 MM could be annotated with eggNOG-mapper [43]. The trimmed reads were aligned against the assembled metagenome with Bowtie 2 [44] and predicted genes were quantified with featureCounts [45]. The complete data processing pipeline was implemented in Snakemake [46] and is available on GitHub [47].

### 2.5. Statistical Analyses

Body weight (BW) (kg), dry matter intake (DMI) (kg/d), metabolizable energy intake (ME MJ/d), and energy corrected milk (ECM) (kg/d) were calculated as average over all daily observations. The ECM was calculated according to [48]. Residual energy intake (REI) was calculated over DIM 3–100 period by fitting a multiple linear regression model with ECM output, metabolic body weight (BW^0.75^), and piecewise regressions of BW gain or BW loss on the total metabolizable energy intake (ME MJ/d), as described in detail by [9]. Thereafter, the daily residuals from the prediction equations were defined to be REI. For descriptive analysis, animals were divided into three REI groups, which contained the lowest (L), middle (M), and the highest (H) third of samples based on three equally spaced quantile groups of the original REI values.

General microbiota composition analyses were performed with *microbiome v. 1.12.0* [49] and *phyloseq v. 1.34.0* [50] R-packages. The analyses included only ASVs that had relative abundance above 0.1% in at least half of the samples, reducing the total number of ASV from 14,755 to 80. The R packages were used to calculate and test Simpson–Gini, Simpson evenness diversity parameters (alpha-function), Bray–Curtis dissimilarities (divergence-function), produce non-metric multidimensional scaling (NMDS) ordination (ordinate-function), and composition plots (plot_composition). Permutational multivariate analysis of variance was done using adonis-function in *vegan v.2.5.7* R-package [51] to assess multivariate differentiation between years and between three REI groups. Differentiation between groups in diversity estimates were done with non-parametric Kruskal–Wallis and, where group differences existed, further explored with pairwise Wilcoxon tests. Correlation tests were performed as two-sided test using cor.test function in the standard R stats package. Phylogenetic UPGMA tree was based on Jukes and Cantor [52] distance as in *ape v. 5.7* [53] and tested using adonis-function in *vegan*, as above.

Predictor engineering. The machine learning model was based on two types of summary variables. First, so-called archetype components were estimated by generalized low rank model (GLRM) which is a generalization of PCA and matrix factorization [54]. Estimation was performed with *h2o v.1.5.2.1* R-package [55]. GLRM Y matrix rows, or archetypal traits, represent microbial community profiles. GRLM X matrix quantifies the presence of these archetypal traits in the samples. The data can be reconstructed from the matrix product XY. The Y matrix was constrained to non-negative values. Limiting to the components explaining more than 1/n(ASV) share resulted in using 26 GLRM archetype components explaining 65% of the variance in ASV frequencies. Components were included after exponential transformation with *caret v.6.0.90* [56]. In addition, Gini–Simpson and Simpson evenness estimates were included as a second type of summary variable. All predictors were centered and scaled before model training.

Extreme gradient boosting modeling. Extreme gradient boosting is an ensemble machine learning method based on a series of decision trees [57] and was performed using *xgboost v.1.5.2.1* R-package [58]. For this, data was randomly split in 8:1:1 proportion to training, validation, and test data sets. Initial tuning assessing tree depths from 1 to 7 with learning rate 0.1 and up to 10 rounds of boosting was done to find the approximate tree depth. Subsequently, two rounds of sequential tuning were done for learning rate (0.01–0.4), predictor subsampling (0.3–1), tree depth (2–3), sample subsampling (0.3–1), and minimum number of samples per node (1–4), running 1000 boosting rounds and choosing the parameter based on convergence and the lowest root mean square error in validation data. The final model hyperparameters were 26 rounds of trees having a depth of two and at least two samples per node. Moreover, a learning rate of 0.05 was used and the trees were constructed without subsampling among the predictors using only a random 80% subsample of the training set for the generation of a tree. The final model was constructed with the xgboost implementation in *caret v.6.0.90* [56], using 100 repeated cross validations.

Prediction interpreters. The predictor impact was analyzed using sample-wise Shapley additive explanation estimates (SHAP) and global accumulated local effects (ALE) estimates. SHAP values refer to the additive contribution of each predictor trait to the sample prediction and the predictor-wise mean of the absolute values is a way to evaluate their importance. Estimation was done using *SHAPforxgboost v.0.1.1* [59]. The ALE profiles, which give the global main effect for the predictors, were obtained using *ALEPlot v.1.1* R-package [60]. A conditional inference tree model was fitted on the machine learning model and the data using TreeSurrogate-function in *iml v. 0.10.1* [61]. The constructed surrogate tree model and comparison between the full machine learning model and surrogate model was plotted using *ggparty v. 1.0.0* [62].

Network analysis. REI-associated ASV variation was analyzed based on reconstructed data. Reconstruction was done by matrix multiplication of GLRM matrices including only the five strongly REI associated archetype components. Sparse Gaussian graphical model network tuned with extended Bayesian information criterion was drawn using the scaled data with EBICglasso-function in *qgraph v.1.9.2* [63]. The network was plotted with *ggraph v.2.0.5* [64], which internally used *igraph v.1.2.11* [65]. igraph was used in calculation of hub scores. Network node sizes in the fr-layout network were proportional to taxon hub scores. Coefficients of association between ASV frequency and REI values were estimated by linear regression with offset equal to the mean value. Network modules were determined using cutreeDynamic-function in *dynamicTreeCut v.1.63* [66]. The input tree was UPGMA-tree based on partial correlation distance (1—partial correlation). The minimum number of ASVs per module was set to 10, and deepSplit parameter to 2. Cluster correlation with REI coefficients was assessed using Pearson correlation and *p*-values were estimated using corPvalueStudent in *WGCNA v.1.70-3* [67,68]. Descriptive cluster names were based on the taxa of the nodes with the highest hub score values, where possible. Differentiation between clusters in archetype component values were done with non-parametric Kruskal–Wallis and Wilcoxon tests.

Shotgun-metagenome data processing. The shotgun-metagenome gene quantification and annotation post-processing was done in R4.2.1 supported by the R-package *data.table* [69]. The downstream analysis was based on highly abundant predicted genes. Highly abundant genes were determined by applying a trimmed mean of M-values normalization (TMM) to the data and filtering out genes with a total sum of standardized values smaller than the 95% quantile across the total sum of all abundances. In total, 1.2 MM filtered genes were evaluated for differential abundances using a generalized Mann–Whitney test for directional alternatives as described by Fischer et al. [70] and implemented in the R-package *gMWT* [71]. Let $F_{L}$, $F_{M}$, and $F_{H}$ be the cumulative distribution functions of abundances for a specific gene g (double index omitted for simplicity) for the groups L-, M-, and H-REI, then, the underlying directional testing problem is $H_{0}:F_{L}=F_{M}=F_{H}$ vs. $H_{1}:F_{L}\leq_{st}F_{M}\leq_{st}F_{H}$ and $H_{2}:F_{L}\leq_{st}F_{M}\leq_{st}F_{H}$ with $\leq_{st}$ being the stochastic ordering that is not equal for at least one comparison. Directional alternatives were preferred for the shotgun-metagenome data analysis over a global test, for example, Kruskal–Wallis, as directional alternatives reassemble the underlying assumption better and avoid umbrella-type results, where intermediate samples would have highest/lowest abundance values. The false discovery rate (FDR) of those tests was controlled with the Benjamini–Hochberg method. Carbohydrate-active enzyme (CAZy) classes as well as the Kyoto Encyclopedia of Genes and Genomes (KEGG) ortholog abundances per sample and group were summarized by summing over the abundances of the corresponding identified genes. For the KEGG pathway enrichment analysis a dataset of all KEGG orthologs (KO) per pathway was created and assessed for differences in KO abundances between the groups using the generalized Mann–Whitney test for directional alternatives. Then, a one-sided Fisher exact test was used to assess for an enrichment of significant differentially abundant KOs (*p*-value ≤ 0.1) for each pathway. For this, a 2 × 2 contingency table was created for each pathway, indicating the amount of significant and non-significant KO within a pathway gene-set as well as the overall KO universe. The KO universe is here the total set of detected KO that are also present in any pathway. Based on the total number of (non)-significant KO the hypergeometric distribution was used to calculate the probability to obtain by chance at least the number of observed significant KO in the pathway gene-set. This probability is then the associated *p*-value for the enrichment test. Further, the enrichment tests were adjusted for multiple testing, by using an FDR of 0.05. The required functions for testing and visualization are implemented in the R-package *GenomicTools* [72]. Term-gene graph was constructed in *pathfindR v.1.6.4* [73].

## 3. Results

### 3.1. REI Association with VFA, Alpha, and Beta Diversities

In total, 87 Nordic Red primiparous dairy cows with REI estimates ranging between −33.4 and 21.3 were included in this study. Kruskal–Wallis test indicated year differences and Wilcoxon test indicated the 2019 samples to have significantly larger values (Appendix A). The REI group differences were assessed in the same way. The average total VFA was 109 mmol/L and varied from 87 to 143 mmol/L. Of the individual VFAs, acetate was significantly higher (*p* < 0.01), while propionate and isovalerate were significantly lower (*p* = 0.02) in H-REI (less efficient) compared to L-REI (efficient) groups (Table 1, Appendix A). There were significant differences in concentrations of five individual VFAs between the years, but not in total VFA concentration (Appendix A).

Alpha diversity, evaluated as Gini–Simpson diversity index, ranged from 0.951 to 0.986 and Simpson evenness from 0.40 to 0.90. There were indicatively positive correlations between REI and Gini–Simpson diversity or Simpson evenness with correlation 0.14 (*p* = 0.2) or 0.19 (*p* = 0.07), respectively. There was non-significant negative correlation (−0.14, *p* = 0.2) between REI and beta diversity.

Rumen microbiota uniformity across the years and REI groups was studied using permutational multivariate analysis of variance. There was differentiation between the years (*p* < 0.002) but not between REI groups (*p* = 0.35). A more detailed analysis was done to understand this observation (Table 1, Appendix A). Among the years, there were significant differences in Simpson evenness (*p* = 0.03) and beta diversity (*p* = 0.006) but not in Gini–Simpson diversity. Among the REI groups, there were no differences in either alpha diversity measure, but L-REI and H-REI groups had significantly higher beta diversity than the M-REI group. The larger dispersion among L-REI and H-REI groups was reflected in the contour of REI values using NMDS ordination coordinates (Figure 1) as presence of the most extreme REI peaks and valleys further from the center. The 2019 samples clustered together agreeing with the observed lower beta diversity. The results indicated that there was significant multivariate differentiation between the years, and it was due to differences in beta diversity.

### 3.2. Predicting REI from Rumen Microbiota

Machine learning model predicting REI proved the link to rumen microbiota composition. The final machine learning model predictions for the training, validation, and test data had strong correlation with the phenotypically estimated (0.88, 0.60, and 0.55, respectively, see Appendix A) and warranted close inspection of the model.

The predictor importance estimates from the SHAP analysis suggested archetype components (Archs) 4, 13, 5, 1, and 24 to have the strongest impacts on predictions, while the importance of the remaining predictors decreased more gradually. Simpson evenness and Gini–Simpson diversity indices ranked 11th and 16th among the 28 predictors (Appendix A), suggesting that several specific ASV assemblages were more important than the overall ASV diversity (Appendix A). ALE plot implied the main effects happen as REI changes within one standard deviation from the mean transformed Arch value (Figure 2). Moreover, the stronger effects occur on positive component values, which biologically means presence of a specific minor microbial sub-community. Only Arch1 and Arch12 showed opposite patterns, suggesting a larger deviation from the mean due to the loss of a common microbial subcommunity. The ALE importance calculated as the differences on minimum and maximum value suggested higher impacts for Archs 4, 5, 13, 24, 9, 12, 1, 20, and 2, with Simpson evenness and Gini–Simpson diversity ranked 10th and 23rd. The predictor importance ranks were similar in both evaluation methods.

A simplified inference tree model was used to capture the main REI associations and was composed of a set of five sample classifying criteria (Figure 3). The coefficient of determination between this simple model and the full model was 0.61. These five archetype components provided the most condensed model explaining the main associations.

### 3.3. Interpretation of the Association Results

The surrogate model results were used to explore the underlying biology from microbial taxonomic data. At genus level, the ASVs were affiliated with 11 genera, among which *Prevotella*, *Succinovibrionaceae UCG-002*, and *Lachnospiraceae NK3A20* group were the most abundant in all studied animals. In particular, the *Prevotella* and *Succinovibrionaceae UCG-002* abundances varied strongly, but independently from REI. Despite ASV level data being associated with REI variation, the genus level taxonomy did not reveal simple rumen microbiota association with REI estimates (Figure 4A). The GLRM analysis enabled reconstructing data based on the important REI associated components only. The reconstructed composition appears more homogenous than the original data but did not indicate simple genus level association to REI either (Figure 4B).

As the next step the REI-associated variation at ASV level was explored and patterns of association between clusters of ASVs and REI phenotype were observed. Co-occurrence network (Figure 5) revealed a sparse network containing only 12% of the possible connections, and the partial correlations were nearly exclusively positive, with a single exception for a *Prevotella*-*Paraprevotella* pair. The co-occurrence module search identified five ASV clusters that differ in their association to REI phenotype. One cluster was central in the network containing the four highest ASV-hub scores and was named Central cluster, other clusters were named according to the taxonomy of ASVs with the highest hub score values. The clusters were also significantly associated with archetype clusters (Appendix A). Increased abundances in *Methanobrevibacter* cluster (REI correlation 0.48, *p* < 0.01, associated to Arch4) and *Prevotella-Succinivibrionaceae* cluster (0.26, *p* = 0.02, Arch1) were associated with H-REI, while higher proportion of *Prevotellaceae* cluster (−0.35, *p* < 0.01, Arch12) and *Central* cluster (−0.33, *p* < 0.01, Arch2) were associated with L-REI. *Ruminococcaceae-Lachnospiraceae* cluster (Arch24) did not have an independent main effect on REI. Overall, the L-REI cows were beneficially associated with more networked bacteria in the rumen. The hub taxa in the network contributed towards feed efficiency (REI correlation −0.43, *p* < 0.0001) between hub score and REI coefficient.

Most genera spanned several co-occurrence clusters. Therefore, a phylogenetic tree was created to explore the association between phylogenetic distances between ASVs and co-occurrence clusters (Appendix A). Analysis demonstrated that phylogenetic distances were not independent of the co-occurrence clusters (*p* = 0.002) and particularly *Succinivibrionaceae* and *Prevotella* showed some evidence of association between phylogeny and functionality above the ASV-level. As an example, *Prevotella*-affiliated ASVs in the *Methanobrevibacter* co-occurrence cluster appear to be limited within a phylogenetic subclade.

### 3.4. Differences in Microbial Functional Capacities between H-REI and L-REI Groups

Of all annotated genes, 2.5% (601,708) were identified to code for carbohydrate-active enzymes (CAZymes). In total, 71 CAZy families were observed in this data set. Among them, 29 were affiliated with Glycoside hydrolases (GH), followed by 19 Glycosyl transferases (GT), 2 Carbohydrate esterases (CE), and 2 Polysaccharide lyases (PL), respectively. There were also 19 clusters, represented by more than one CAZy family. Among the CAZymes, several families of cellulases (GH5, GH6, GH9, GH95), hemicellulases (GH8, GH23, GH26, GH28, GH43), debranching enzymes (GH33, GH51, GH77), and a large group of oligosaccharides degrading enzymes (GH3, GH13, GH18, GH20, GH29, GH31, GH32, GH38, GH57, GH94) were observed. Among them, GH13, GH31, and GH3 were the most abundant families. GTs were the second most prominent CAZymes group with enzymes belonging to 19 families, among which GT5, GT51, GT20, GT35, GT36, GT2, and GT4 were the more abundant. The minor groups were CEs and PLs, with CE1 and CE10 as well as PL8 and PL11 families present in this data set (Appendix A). From the total set, abundances of 25 CAZy families were significantly (FDR < 0.05) different between H- and L-REI groups. These differences were related to the significantly higher abundance of carbohydrate-degrading enzymes from the GH3, CBM48-GH13, GH31-GH97, GH43, GH57, GH8 families and higher abundance of several GT families (GT51, GT2-GT4, GT19) in L-REI cows. On the other hand, H-REI group had significantly higher enrichment of GT66 involved in glycan biosynthesis, GT5 and GT39 involved in starch and sucrose metabolism, or GT2 involved in lipopolysaccharide biosynthesis (Figure 6), suggesting that L-REI animals demonstrated higher capability to degrade complex carbohydrates, while H-REI cows showed higher capacity for processing polysaccharides.

In total, 37.1 MM genes were predicted from the shotgun-metagenome. Of those, 35.3% were not present in the eggNOG database and were removed from further functional analysis. The remaining 24 MM were filtered based on their overall abundance level, leaving a set of 1.2 MM highly abundant genes for the downstream analysis. In total, 18,084 genes of this set were significantly higher in abundance (DA) in the H-REI group and 8729 in the L-REI cows. A principal component analysis based on the DA genes separated the L- and H-REI groups (Appendix A). In total, 478 KEGG pathways were identified in the data set, which belonged to four first level KEGG functional categories. Based on the read abundances, 54–60% belonged to the “metabolism”, 14–18% to “organismal systems”, while 8–10% were affiliated with “genetic information processing”, “environmental information processing”, and “cellular processes”, respectively (Appendix A). The pathway completeness analysis was performed based on the coverage of KOs, associated with the pathways, and demonstrated 52.4% (IQR: 40.3%–67.0%) completeness (Appendix A). KO abundances for all the 12,946 identified KOs were then calculated by summing over all TMM normalized gene abundances associated with individual KOs. In total, 3882 (592) KOs were more abundant in the H-REI (L-REI) group.

In the KEGG pathway enrichment analysis there were 29 pathways more abundant in H-REI cows and 3 pathways enriched in the L-REI group. All these pathways belonged to the first level KEGG functional categories “genetic information processing”, “environmental information processing”, and “cellular processes”. Among “genetic information processing”, at the second level of KEGG categories, the H-REI cows were enriched in metabolic pathways related to “transcription” processes (03040 Spliceosome), “translation” processes (03010 Ribosome, 03013 Nucleocytoplasmic transport, 03015 mRNA surveillance pathway and 03008 Ribosome biogenesis in eucaryotes), and “folding, sorting, and degradation” (04141 Protein processing in endoplasmic reticulum, 04120 Ubiquitin mediated proteolysis, 03050 Proteasome). Among the “cellular processes”, the H-REI group was enriched in “cell growth and death” (04110 cell cycle, 04111 cell cycle-yeast, 04113 meiosis-yeast, 04114 oocyte meiosis, 04218 cellular senescence) and “transport and catabolism” (04144 endocytosis, 04142 lysosome, 04137 mitophagy-animal) pathways. On the other hand, under the “cellular processes” category, the L-REI group was enriched in “cell motility” (02030 bacterial chemotaxis, 02040 flagellar assembly) pathways and under “environmental information processing” category in “signal transduction” (02020 two-component system) metabolic pathway (Figure 7). The completeness of these pathways as well as KO-wise ratio values between the L-REI and H-REI groups are visualized in Appendix A.

## 4. Discussion

In this study, a cohort of 87 primiparous Nordic Red dairy cows was used to investigate if rumen microbiome composition and function can be used as indicators for animal feed efficiency trait. By using gradient boosting, multiple microbial networks were identified that were positively or negatively associated with the efficiency phenotype of Nordic Red dairy cows, and they were used for predicting the phenotype. This suggests that microbial consortia, not individual taxa, significantly influence feed efficiency. Further, microbial metagenomic analysis demonstrated interdependence between microbial functional potential and dairy cow efficiency.

The microorganisms in the ecosystem can be defined as generalists or specialists, with generalists having a broad functional capability to adapt to wide ecological space, whereas specialists are restricted to a narrow range of resources and conditions [74]. It is recognized that different functional and metabolic capabilities, conducted by networks of such microorganisms, are needed to occupy distinct ecological niches in the rumen but it is their collective interaction that will influence the host’s digestive processes and energy production. In this study, several ASVs were observed in multiple sub-networks and could represent generalists of rumen ecosystem. For example, *Lachnospiraceae* NK3A20 group, *Ruminococcaceae* UCG-001, *Christensenellaceae* R-7 group, and *Acetitomaculum*, all representatives of Firmicutes, were detected in both H- and L-REI related networks. Similarly, many *Prevotella*-affiliated ASVs, most of them not defined deeper than at genus level, occurred across the networks. Some general roles have been reported for these genera. *Lachnospiraceae* NK3A20 group is most phylogenetically related to *Butyrivibrio* [75], a bacterial genus that can degrade xylan found in plant cell walls and produce butyrate [76]. *Ruminococcaceae* family consists of mainly fibrolytic organisms, but some species can utilize starch [77]. *Christensenellaceae* R-7 group can degrade carbohydrates, amino acids, and carboxylic acids, producing acetate and a small amount of butyrate [78], while *Acetitomaculum* can utilize hydrogen to reduce carbon dioxide to form acetate [79]. The presence of such microorganisms in several networks may show a common interdependence between primary and secondary metabolizers, where end or intermediate activities of one microorganism produces substrates to be utilized by another member of the ecosystem.

In contrast, the ASVs classified under *Prevotella bryantii, Paraprevotella*, and *Rikenellaceae* RC9 gut group, all representatives of Bacteroidota, and *Succiniclasticum* (Firmicutes), were observed only within L-REI associated networks and could indicate more specialized functions inside the network. Members of *Prevotellaceae* have been observed to have both positive and negative association with feed efficiency [18,80] or methane [81]. *Prevotella* strains are genetically and metabolically diverse [82], capable of utilizing starch, protein, hemicellulose, pectin to generate acetate, succinate, and propionate. This variation in substrate preferences allows them to occupy distinct niches within the rumen ecosystem and form unique interactions with other microorganisms, impacting animal performance in an either positive or negative manner. *Paraprevotella*, known to be isolated from human feces, can produce succinate and acetate as end products of glucose metabolism [83]. *Succiniclasticum ruminis* is a common inhabitant of the rumen and can ferment succinate to propionate without a requirement for other carbohydrates [84]. Myer et al. [85] observed higher abundance of *Succiniclasticum* spp. among efficient steers, while Clemmons et al. [86] detected a greater concentration of succinate/methylmalonate in efficient beef cattle. *Rikenellaceae* RC9 gut group are members of the *Rikenellaceae* family which can produce propionate, acetate, and/or succinate as a fermentation product. The *Rikenellaceae* RC9 gut group was positively correlated with growth performance in beef [14]. Although more evidence is needed, microbial interactions involved in propionate synthesis through the succinate pathway could be linked with feed efficiency in Nordic Red dairy cows. Shabat et al. [19] observed the opposite, where the succinate pathway was associated with non-efficient cows, but the acrylate pathway was enriched in efficient animals. The differences between the studies may be due to the higher proportion of concentrate in the diet by [19] that could have led to a higher abundance of lactic acid utilizing bacteria in the rumen as compared to the diet in this study, indicating that functional potential of rumen microbiome can serve as a proxy for feed efficiency only among herds managed under similar dietary conditions.

Another microbial network that was associated with a disadvantageous impact on REI had *Methanobrevibacter ruminantium* as a hub microbe in the cluster. Methane production is considered a significant energy loss and feed efficient animals have been observed to produce less methane, but the results are not conclusive [23,87]. Nevertheless, higher abundance of *Methanobrevibacter* was detected in inefficient cows [19,28], while different *Methanobrevibacter* species have been enriched in efficient and inefficient sheep [20]. It has been demonstrated that no direct correlation exists between the abundance of archaea and methane emissions. We hypothesize that a significantly higher concentration of acetate and higher abundance of archaea in H-REI cows indicate higher concentration of H_2_ in the rumen available for utilization. Redirecting H_2_ uptake for propionate synthesis, as visible in L-REI cows, or presence of alternative H_2_ consumption pathways, could be linked with more efficient energy utilization and at least partially explain the differences between efficient and less efficient animals in this study.

The use of microbial network analysis as a multivariate biomarker for animal performance efficiency studies has its own limitations. Previous research demonstrated the importance of the diet on the formation of microbial networks, where changing forage to concentrate ratio or addition of oil to the diet influences which microbial taxa function as main information hubs within the networks [11]. This suggests that microbial interactions within the rumen ecosystem are not static but dynamic and will be affected by the diurnal rhythm, feeding regime, and dietary composition, therefore, making generalized conclusions from independent studies may be difficult.

Ruminants’ ability to deconstruct structural plant polysaccharides into fermentable sugars is dependent on the large diversity of polysaccharide degrading enzymes, produced by the rumen microorganisms. Therefore, we explored if animals managed similarly and fed the same diet but showing differences in residual energy efficiency express differences in CAZymes composition. The most abundant GH13 and GH31 families were not significantly different between the L- and H-REI groups, while more differences were observed among less abundant CAZy families. L-REI cows had a higher abundance of 1,4-alpha-glucan branching enzyme [EC:2.4.1.18] (CBM48-GH13) that converts amylose into amylopectin and, based on eggNOG database, was produced by bacteria, fungi, and protozoa. In addition, they had more of beta-glucosidase [EC:3.2.1.21] (GH3), arabinan endo-1,5-alpha-L-arabinosidase [EC:3.2.1.99] (GH43), alpha-glucosidase [EC:3.2.1.20], glucan 1,4-alpha-glucosidase [EC:3.2.1.3] (GH31-GH97), and oligosaccharide reducing-end xylanase [EC:3.2.1.156] (GH8) all involved in carbohydrate metabolism and transport, and stemming mainly from Bacteroidota. Our observations suggest that both prokaryotic and eukaryotic hosts contributed to the higher abundance of plant degrading enzymes in L-REI cows.

Contrary to L-REI cows, H-REI group was enriched in several GTs. Similar enrichment of GTs has been observed among feed inefficient beef cattle [22]. Glycosyltransferases are required for the transfer of sugars to a variety of important biomolecules, including glycans, lipids, peptides, and small molecules. Among the more abundant GTs in H-REI cows was starch synthase [EC:2.4.1.21] (GT5), involved in glycogen synthesis. In the environment, where flow of nutrients is not stable, many microorganisms accumulate glycogen as carbon and energy reserve and utilize it when environmental glucose supply is temporarily depleted [88]. Capability to synthesize and utilize glycogen supports microbial cells undergoing various physiological transitions, e.g., between planktonic and biofilm lifestyles or enable them to occupy more diverse niches [89]. In addition, the H-REI group was also enriched in CDP-glycerol glycerophosphotransferase [EC:2.7.8.12] (GT2) involved in the biosynthesis of poly glycerol phosphate teichoic acids in bacterial cell walls, dolichyl-diphosphooligosaccharide protein glycosyltransferase [EC:2.4.99.18] (GT66) involved in glycoprotein synthesis and dolichyl-phosphate-mannose-protein mannosyltransferase [EC:2.4.1.109] (GT39) that transfers mannosyl residues to the hydroxy group of serine or threonine residues, producing cell-wall mannoproteins. This suggests that in H-REI animals more of the microbial functional capacity was directed towards accumulation of energy sources and cell destruction/multiplication processes as compared to the rumen of efficient cows.

At the overall KEGG pathway level, no differences between L- and H-REI groups were detected in the predominant “metabolism” category, which is affiliated with metabolic processes related to carbohydrate, lipid, amino acid, or nucleotide metabolism, among others. Contrary to these results, enrichment of several pathways related to amino acid metabolisms was observed among inefficient beef steers [22], while some genes from the carbohydrate metabolism pathway were identified as predictive for feed efficiency in beef [90]. Considering that the concentrate proportion in the diet of beef cattle is generally higher compared to the diet of dairy cows, these observations could suggest that rumen microbiome functions, differentiating animals by feed efficiency phenotype, are diet dependent.

Among other KEGG categories, we observed a higher number of enriched KEGG pathways in H-REI cows as compared to L-REI animals, supporting previous observations in Holstein cows [19] and beef [22] that less efficient animals have rumen microbiome involved in more diverse metabolic activities. On the other hand, enrichment of various pathways related to “transcription”, “translation”, “folding, sorting, and degradation”, or “signal transduction” and “cell growth and death” processes in H-REI cows could suggest processes leading to more active microbial cell proliferation among prokaryotes but also eukaryotes in rumen of less efficient animals. Higher expression levels of “translation” and “transcription” pathways were also detected in less efficient beef metatranscriptome data [13] and support our hypothesis. In addition, H-REI cows were enriched in “Phosphatidylinositol signaling system” (ko04070) (Appendix A), as well as pathways related to autophagy (ko04138, ko04136), “endocytosis” (ko04144), and “lysosome” (ko04142) which are related to the processes in eukaryotic cells. Phosphoinositides are cellular phospholipids with a broad range of functions. They regulate ion channels, pumps, and transporters and control processes related to cell proliferation, survival, vesicle trafficking, membrane dynamics, autophagy, or cell division [91]. Rumen ciliate protozoa are predators [92] and higher abundance of KOs related to processes involving endocytosis, formation of phagolysosomes, and autophagy in this study, could be an indicator of higher predatory activity of ciliate protozoa, engulfment, and recycling of cellular material in the rumen of H-REI cows. Taken together, these observations suggest more emphasis on growth of both prokaryotes and eukaryotes, due to possibly higher availability of simple nutrients, at least transiently, in the ruminal environment of the H-REI cows. In addition, differential abundance of eukaryote-related metabolic processes highlights the need to assess the role of ruminal eukaryotes in understanding performance traits of animals.

The limited number of enriched KEGG pathways in L-REI cows had emphasis on bacterial environmental sensing and motility (“ko02030 bacterial chemotaxis”, “ko02040 flagellar assembly”). Similarly, motility processes were enhanced also in efficient beef [90]. Bacterial chemotaxis plays a key role in many biological processes. It is the process by which cells can sense chemical gradients in their environment and move towards more favorable conditions through the motility of the flagella, which helps in search for nutrients but also for their survival. Chemotaxis is important in biofilm formation, quorum sensing, bacterial pathogenesis, or host infection [93]. To sense extracellular environment triggers, bacteria utilize stimulus-response coupling signaling systems known as two-component systems (TCS). Each TCS consists of two proteins, a sensor, and a response regulator [94] and their interaction triggers changes in bacterial gene expression. Although the “ko02020 two-component system” pathway was enriched in L-REI cows, there are hundreds of such systems in bacteria and over-generalization of pathway importance should be avoided. Nevertheless, in this study we observed examples of enrichment of different quorum sensing TCS systems in L-REI and H-REI cows (Appendix A). The L-REI group had higher abundance of LytTR family sensory signal transduction system, found primarily in Gram-positive bacteria, where they are involved in regulating a variety of cellular processes, including virulence, competence, or bacteriocin production [95]. The H-REI cows were enriched in the LuxR family, detected primarily in Gram-negative species involved in regulating quorum sensing and the expression of genes involved in virulence, biofilm formation, and antibiotic resistance. Taken together, these observations suggest that the rumen of H-REI and L-REI form distinct types of niches with microorganisms facing different challenges. Moreover, the results suggest more emphasis should be based on between-kingdom interactions. The differences can be speculated to be influenced by cow feeding or rumination behavior or the formation of diverse types of microbial communities by change or through interaction with the host.

Machine learning algorithms refer to various data analysis methods often emphasizing prediction rather than statistical significance. Despite the potential advantages of this approach, predicting phenotypes in in vivo feed efficiency studies has been challenging due to difficulties in collecting large enough data sets (however, see [26,28,86]). In microbiome data, the number of predictors often greatly exceeds the number of samples, which can lead to model overfitting and limit cross-study comparison. Predictor number can be reduced using only a coreset of more abundant microbes occurring widely, as this is not expected to cause significant biases or false outcomes [96], although this is unlikely to be sufficient alone. Analyses using higher taxonomic aggregates have been observed to loose predictive power [30] agreeing with current results. Hierarchical taxonomy-aware feature engineering [97] has been suggested as a compromise where lower-level information is combined to higher-level aggregates when it reduces the number of correlating features; however, the method did not significantly reduce number of predictors currently. The current study supports reducing the predictor number using more general unsupervised multivariate methods encapsulating the largest amount of statistical variance or information. Most machine learning methods are expected to suffer from high dimensionality; in the current study it was seen in all the tested 20 supervised methods ranging from clustering- and regression-based methods to neural nets during initial method selection. Method comparison agreed with the common suggestion that for tabular data random forests give acceptable quick results and gradient boosting methods give more accurate results after careful model tuning. The initial correlations between phenotypic estimates and predictions in the validation data were only around 0.3 for the better performing methods, and the subsequent improvement to 0.6 for the validation data set was due to hyperparameter adjustments to decrease overfitting, emphasizing the importance of model tuning in gradient boosting. Despite successful phenotype rank prediction for the test set (data not used in model training or in hypervariable tuning), the study is limited to discovery research as no fully independent data was used for verification.

## 5. Conclusions

In a cohort of 87 Nordic Red dairy cows, it was demonstrated that rumen liquid microbiota can be used to predict REI phenotype using a modern ensemble machine learning model. With associated methods explaining the model predictions, machine learning helped reveal the underlying biology, and pointed towards the importance of specific microbial interacting consortia instead of general diversity or individual taxa. It was also demonstrated that rumen microbiome functional potential, evaluated through shotgun-metagenome sequencing, discriminated between efficient and non-efficient animals, but emphasized the need to study both prokaryotic and eukaryotic rumen community interactions for a better understanding of their association with host production traits. It must be pointed out that association does not imply causation, and the findings might not be readily transferable to other populations and environmental conditions, therefore, replicating these findings is necessary.

## Figures and Tables

**Figure 1 microorganisms-11-01116-f001:**
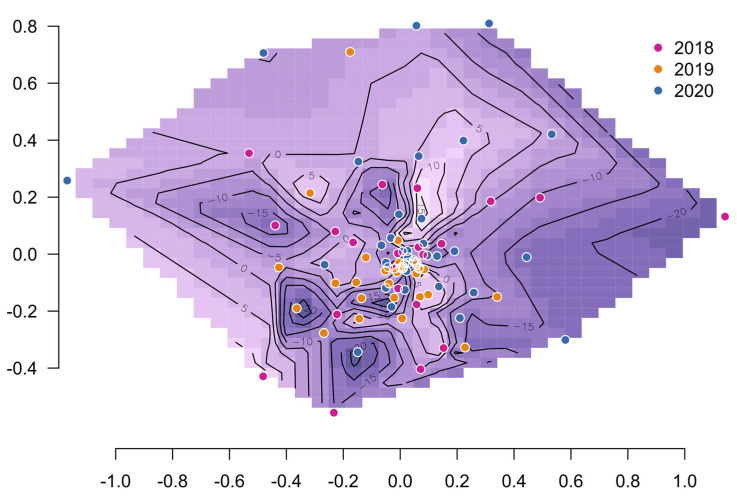
Contour surface of REI estimates (darker areas indicate a low REI neighborhood) on Bray–Curtis distance-based NMDS sample coordinates.

**Figure 2 microorganisms-11-01116-f002:**
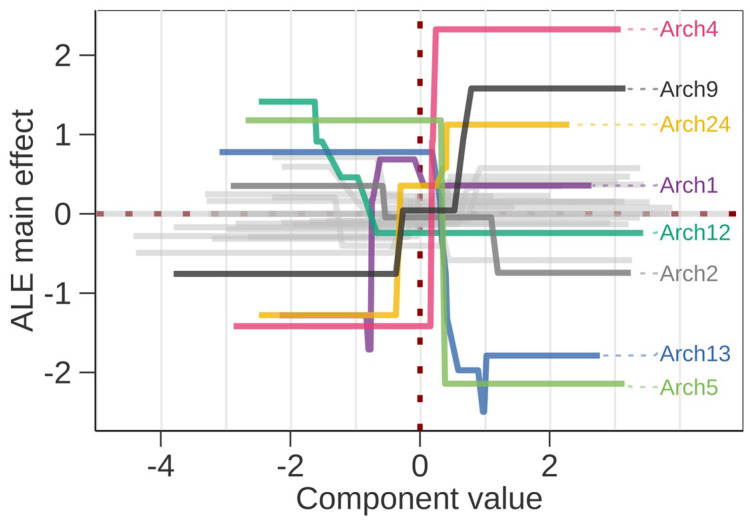
Accumulated local effects plot for all predictors with component values on x-axis and impact on prediction on y-axis. Only the archetype components (Arch) where the difference between maximum and minimum local effect is >1.5 are highlighted by color and labeled. In addition, Arch2 that appears in the surrogate model is highlighted. On each axis the average value is zero and the component values are scaled to unit standard deviation.

**Figure 3 microorganisms-11-01116-f003:**
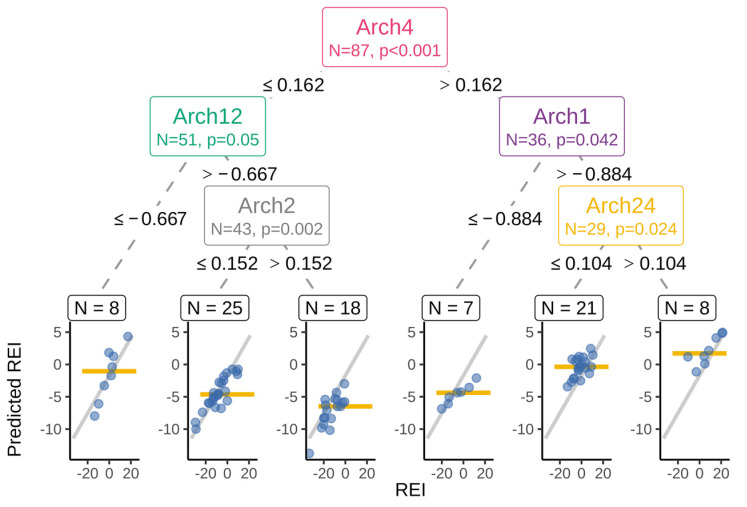
Simplified surrogate model representing the main findings as an inference tree. Archetype component (Arch) name, number of cows, statistical significance test, the split thresholds given at nodes, and the child branches. Leave scatter plots show the original REI (on the x-axis) and the model predicted REI (y-axis) for each animal (blue circle) and the simplified tree prediction for the group as a horizontal line. Gray ascending line is the fitted overall regression line.

**Figure 4 microorganisms-11-01116-f004:**
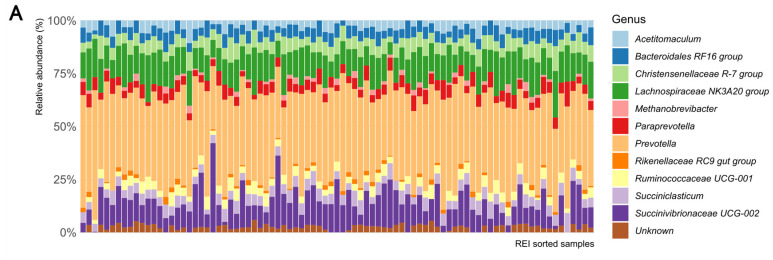
The rumen bacterial composition at genus level of samples ordered according to their REI estimate, indicating (**A**) original composition and (**B**) reconstructed REI associated composition.

**Figure 5 microorganisms-11-01116-f005:**
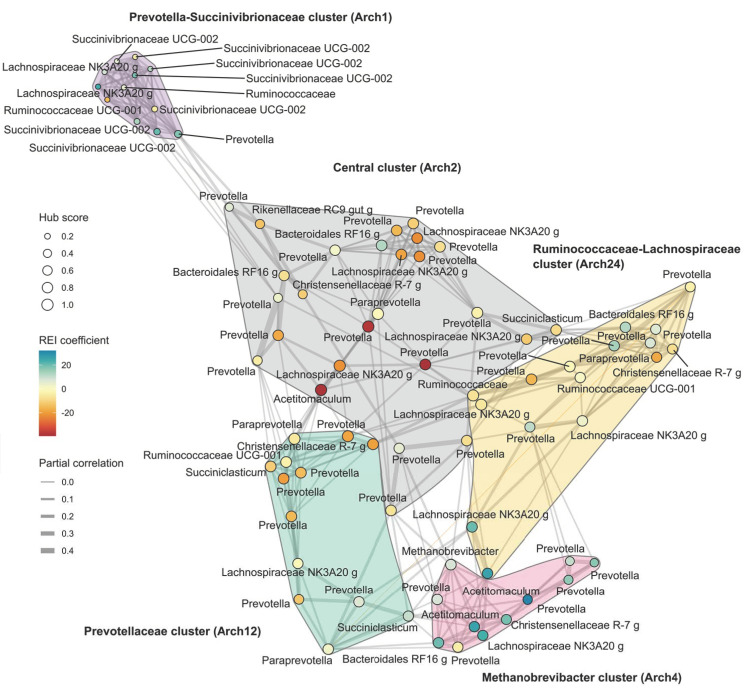
Co-occurrence network based on the reconstructed abundance data. In the network the hub scores and the associations to REI are also indicated. The co-occurrence clusters are named based on their main hub taxa and their archetype component (Arch) association is indicated in parentheses. The single negative correlation is indicated as an orange edge linking the *Paraprevotella* to the *Prevotella*.

**Figure 6 microorganisms-11-01116-f006:**
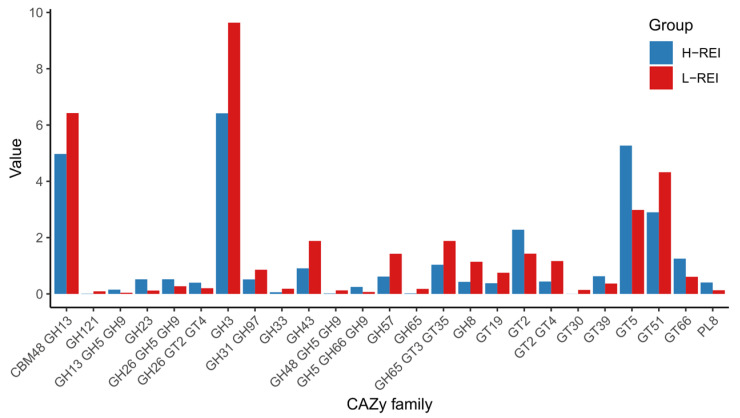
Abundances of CAZy families that were identified as significantly (FDR < 0.05) different between H-REI and L-REI groups.

**Figure 7 microorganisms-11-01116-f007:**
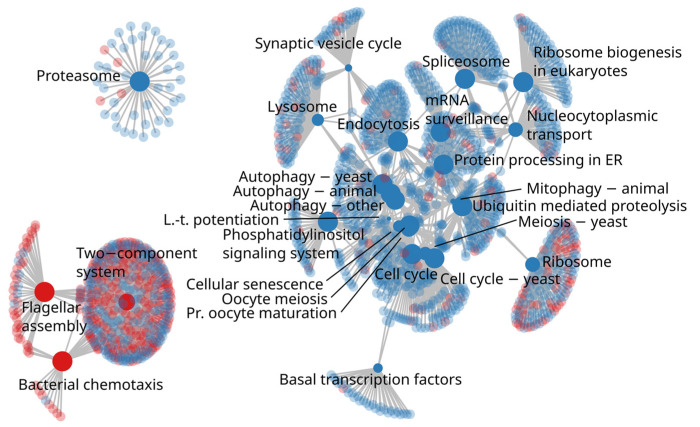
Term-gene graph showing enriched KEGG pathways connected to their orthologs. Red color indicates enriched pathways in L-REI individuals, and blue indicates enrichment in H-REI individuals. The KEGG pathways are solid and the orthologs are semi-transparent, respectively. The term node size reflects the negative logarithm of the enrichment test *p*-value, which ranged from 10^−6^ (e.g., Proteasome) to 0.002 (Long-term potentiation; Mitophagy-animal). Few terms were shortened for visual clarity: Long-term potentiation (L.-t. potentiation), Protein processing in endoplasmic reticulum (Protein processing in ER), mRNA surveillance pathway (mRNA surveillance), and Progesterone-mediated oocyte maturation (Pr. oocyte maturation). Human disease terms were not included.

**Table 1 microorganisms-11-01116-t001:** Animal numbers in groups (*n*), and medians and interquartile ranges across the animals for residual energy intake (*REI*), mean energy corrected milk (*ECM*), mean bodyweight (*BW*), mean dry matter intake (*DMI*), total volatile fatty acids (*tVFA*), acetate (*Acetate*), isovalerate (*Isovalerate*), propionate (*Propionate*), and Gini–Simpson diversity (*Gini–Simpson*), Simpson evenness (*Evenness*), and beta diversity (*Beta*) per group and overall.

	L-REI [−33.4, −9.17]	M-REI (−9.17, −0.9]	H-REI (−0.9, 21.3]	All
n	29	30	28	87
REI *	−14.3 (7.2)	−4.6 (4.8)	5.1 (7.1)	−4.6 (14.5)
ECM	30.6 (4.9)	30.7 (4.1)	29.5 (2.7)	30.26 (4.3)
BW	614 (107)	604 (61)	589 (80)	601 (81)
DMI	20.1 (2.8)	20.6 (1.9)	21.1 (1.5)	20.7 (2.0)
tVFA (mmol/L)	109 (16)	109 (17)	107 (14)	108 (16)
Acetate (mmol/mol)	650 (20) ^B^	655 (24) ^AB^	659 (27) ^A^	654 (24)
Isovalerate (mmol/mol)	10.8 (3.7) ^A^	10.4 (3.3) ^AB^	9 (4.1) ^B^	10.3 (3.5)
Propionate (mmol/mol)	190 (19) ^A^	187 (19) ^AB^	182 (16) ^B^	187 (18.5)
Gini–Simpson	0.975 (0.008)	0.979 (0.008)	0.977 (0.007)	0.978 (0.009)
Evenness	0.800 (0.065)	0.823 (0.057)	0.817 (0.038)	0.817 (0.050)
Beta	0.353 (0.126) ^A^	0.316 (0.103) ^B^	0.339 (0.136) ^A^	0.335 (0.133)

^A,B^ Within a row, medians without a common superscript differ (pairwise Wilcoxon test *p* < 0.05). * Used in setting up REI groups, therefore differentiation among REI groups was not evaluated.

## Data Availability

Raw sequences of the V4 16S rRNA gene region in FASTQ file format for all samples have been deposited in the NCBI SRA repository under the BioProject accession number PRJNA925111. Metagenomics raw sequencing data can be downloaded from the European Nucleotide Archive (ENA) under study accession PRJEB60902.

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
