# Peer review of "Rumen Microbiota Predicts Feed Efficiency of Primiparous Nordic Red Dairy Cows"

_microorganisms, 2023, doi:10.3390/microorganisms11051116_

Round 1

Reviewer 1 Report

Dear authors,

The manuscript is engaging and interesting. There is quality and much effort put in this work, using high-level techniques, well presented, and it will not only give more insight into the association between rumen microbiota and feed efficiency but also be a steppingstone into helping other studies to also work on predicting feed efficiency phenotype through ruminal microbiota for any ruminant species.

Major comments:

Introduction:

I believe there is extensive general information in the introduction. More specifically, in Lines 39-48, and in Lines 54-61. Please try to reform it.

Materials and Methods:

Data regarding the diets are missing. Please if possible, include a table of the composition of the diets as well as their chemical composition.

From an animal science perspective, it would be wise to include even in supplementary material the descriptive statistics of the parameters such as BW, DMI, ME, ECM, BW0.75 and present them per season and/or per group with the number of the animals included in each group (L, M, and H) (more or less something like Table 1).

L167-171. The model needs to be in detail described. Also, can you please explain more comprehensively how the groups were formed? The clustering procedure and selection criteria of the groups L, M, and H need to be described.

Correct me if I am wrong but by my understanding, you divided the efficiency groups based on REI. However, did you choose animals of different diets and seasons for example in a group? If that is so, was this considered in the models? Since we know that both seasonal variation and diet are crucial in modifying rumen microbiota.

Results:

Results are well presented, with high-quality Figures and Tables.

Discussion:

The discussion is extensive yet explanatory and good in that it points out a few limitations.

Minor spelling comments:

Line 23: Please add “an” before inefficient.

Line 97: It is better to use the word rumen than gut.

Line 482: Probably omit “on”.

Line 517: Add “the” before succinate.

Reviewer 2 Report

As a general evaluation the Paper is interesting and important.

The evaluation will be performed section by section to be easy the comprehension.

Introduction

There are a good literature review. However, it is extensive. I'd sugest for this section reduce the content. A good Introduction can be write in a page, delivering the central information and presenting the gaps which will be evaluated in the trial.

Material and Methods

Animals, experimental design, diet

Please provide more info about diet. At least a paragraph describing the proportions and chemical composition (DM, CP, NDF, ADF, TDN). These info is crucial to understand the rumen microorganisms during the different years mainly.

Results

General Consideration: There are some information that should be described in 'Material and Methods' section because it is not results but information about the tunning, and even 'Discussion' section because it is exploring the results discussing about them.

Table 1.

Confusing with a,b and c upper letters to describe differences between years and REI-groups.

Figure 4

Provide inside the figure whats the Figure 4. a and Figure 4. b.

File S1 Figure S9

The variance from PC1 and PC2 was small and observing the plot the separation between the groups is not well defined.

The constant use of supplementary material to presented extra information make the Paper hardish to be read due the constantly changes between files.

Discussion

The 1st paragraph should be removed. There is no discussion inserted in it.

The afirmation inside Material and Methods section: "diet was adjusted based on the stage of lactation and the digestibility of the grass silage" increase the importance to describe better the diet across the periods, which should be discussed and correlated with the  interdependence between primary and secundary metabolizers.

Adjust some citations according to Journal rules.

There are some affirmation inside the 'Discussion' that should be pointed with references due it importance.

Excessive self mention across the Paper - e.g., "we showed", "we demonstrated". Avoid to use "I", "me", "we", replace these for "the study", "this trial".

Conclusion

Your conclusion should answer your hypothesis. Are you doing this in your Conclusion section?

Reviewer 3 Report

In general, the statistical procedures used in the study are confusing. There is a lack of clarity regarding the statistical tests, with authors using different names for the same test (e.g., Mann-Whitney-Test/Wilcoxon). It is also unclear why the Kruskal Wallis or the Mann-Whitney-Wilcoxon tests were chosen for specific situations or comparisons. Additionally, while the authors used non-parametric tests, means and standard deviations were presented, which is an error. Correlation analysis was not described in the Materials and Methods section but was mentioned in the Results section.

I have concerns about the rumen microbiology procedures used in the study. Sampling from a single location within the rumen may not fully reflect the overall bacterial community composition. Moreover, the method used to collect rumen samples via esophageal tubing may bias the sampling towards collecting a higher proportion of rumen fluid, especially in cows fed grass silage and concentrate mix. It is well-established that a large proportion of microbial species in the rumen are particle-associated, and several trials have shown significant differences between solid and liquid rumen fraction bacterial analyses or rumen sampling by oral tube effect (doi:10.1016/j.anaerobe.2011.04.018; doi: 10.1016/j.vetmic.2013.02.013; Henderson et al. 2013. PLOS One 8:e74787). Therefore, the data may not be representative of the rumen bacterial community composition.

 Specific comments:

 Keywords: Avoid using the same words as the title. 

Line 45-50: Please include a reference to support this information.

 Line 53-59: Several references need to be included to support this information.

 Line 116-117: Please explain the "automatic scale" method more clearly.

 Line 128: For how long were the samples kept at 80°C until DNA extraction?

 Line 138-139: Please provide a better description of the clustering process used.

Line 166-167: The number of daily observations used was not described in the Materials and Methods section.

 Line 248-249: Please provide a better explanation of Fisher's exact test, which is based on the hypergeometric distribution.

 Line 256-257: While the Kruskal Wallis test was described in the Results section, other test descriptions (e.g., Wilcoxon) were presented in the table footnote.

 Line 264-265: Correlation tests were not described in the Materials and Methods section.

 Table 1: The format of Table 1 should be improved. The year and REI groups could be presented as columns, while variables such as REI, tVFA, Acetate, etc. could be presented as rows. This would allow for the inclusion of other data, such as propionate and isovalerate, which also differed among groups. As the focus of the manuscript is on the REI groups, the year data could be presented in the supplementary material. Given that the authors used non-parametric tests, data should be presented as median and IQR.

 There are too many abbreviations in the article, and some of them are unnecessary.

The format of references is not uniform. Please review carefully. For example: "Dowle, M.; Srinivasan, A. data.table: Extension of data.frame; 2021."

Round 2

Reviewer 1 Report

The manuscript was adequately revised based on comments and suggestions.

Reviewer 3 Report

The authors have clarified all my concerns and adjusted the paper considering the recommendations of both reviewers.